# Structural Peculiarities of Natural Ballas—Spheroidal Variety of Polycrystalline Diamond

Andrei A. Shiryaev [1,2,*], Felix V. Kaminsky [3], Anton D. Pavlushin [4], Vasily O. Yapaskurt [5], Denis A. Zolotov [6], Alexei A. Averin [1], Olga M. Zhilicheva [2], Maximilian S. Nickolsky [2] and Olga V. Kuznetsova [3]

1 Frumkin Institute of Physical Chemistry and Electrochemistry RAS, Leninsky Prospect 31 bld. 4, 119071 Moscow, Russia; alx.av@yandex.ru
2 Institute of Geology of Ore Deposits, Petrography, Mineralogy and Geochemistry RAS, Staromonetnii per. 35, 119017 Moscow, Russia
3 Vernadsky Institute of Geochemistry and Analytical Chemistry RAS, Kosygina st. 19, 119334 Moscow, Russia; kaminsky@geokhi.ru (F.V.K.)
4 Diamond and Precious Metal Geology Institute Siberian Branch RAS, Lenina st. 39, 677077 Yakutsk, Russia; pavlushin@diamond.ysn.ru
5 Department of Geology, Lomonosov Moscow State University, Leninskie Gory 1, 119991 Moscow, Russia; yvo72@geol.msu.ru
6 FSRC "Crystallography and Photonics" RAS, Leninsky pr. 53, 119333 Moscow, Russia; zolotovden@mail.ru
* Correspondence: shiryaev@phyche.ac.ru or a_shiryaev@mail.ru

**Abstract:** Ballas is a rare polycrystalline diamond variety characterized by a radially oriented internal structure and spheroidal outer shape. The origin of natural ballases remains poorly constrained. We present the results of a comprehensive investigation of two classic ballas diamonds from Brazil. External morphology was studied using SEM, high-resolution 3D optical microscopy, and X-ray tomography. Point and extended defects were examined on polished central plates using infra-red, photo- and cathodoluminescence spectroscopies, and electron back-scattering diffraction; information about nanosized inclusions was inferred from Transmission Electron Microscopy. The results suggest that fibrous diamond crystallites comprising ballas are split with pronounced rotation, causing concentric zoning of the samples. Pervasive feather-like luminescing structural features envelop single crystalline domains and most likely represent fibers with non-crystallographic branching. These features are enriched in N3 point defects. Twinning is not common. The nitrogen content of the studied samples reaches 700 at.ppm; its concentration gradually increases from the center to the rim. Annealing of the ballases took place at relatively high temperatures of 1125–1250 °C; the annealing continued even when the samples were fully grown, as suggested by the presence of the H4 nitrogen-related defects in the outer rim. Presumably, the ballas diamond variety was formed at high supersaturation but in conditions favoring a small growth kinetic coefficient. The carbon isotopic composition of the studied ballases ($\delta^{13}$C = −5.42, −7.11‰) belongs to the main mode of mantle-derived diamonds.

**Keywords:** diamond; ballas; split crystals

## 1. Introduction

Ballas is a relatively rare variety of polycrystalline diamond, characterized by a radial fibrous structure (variety VI in classification by Orlov [1]). It is often spherical and possesses peculiar surface features. Ballas and ballas-like diamonds are found in placer deposits of Brazil [2], in particular in Bahia and Minas Gerais states, in several localities in Russia—Ural mountains [3], the Sayan region [4], and Yakutia [5,6]. Ballas diamonds were also found in the Premier and Orapa kimberlite pipes in southern Africa [7] and in the Sytykanskaya pipe in Yakutia [5]. These diamonds are characterized by a rather broad size distribution of comprising crystallites varying from sub-micron grains to single crystalline domains

reaching 0.2–0.3 mm. Common nitrogen defects such as A, B, N3, and associated platelets are present [8,9]. Only a few analyses of classic ballases have been reported showing enrichment in light carbon isotopes: $\delta^{13}$C VPDB from −14.2 to −21.4‰ [10]; for ballas-like diamonds (i.e., spheroidal diamonds, aggregates of radially-arranged crystallites) values between −5.35 and −11.2‰ were obtained [11]. Several studies addressed the morphology and internal structure of natural ballases [7,9,12–16]. It is shown that the degree of the radial-fibrous internal structure development varies significantly [9,11,15,16]. A genetic model relating carbon supersaturation in a growth medium with morphology and structure of ballases and ballas-like diamonds was proposed in [16]. It is supposed that the growth of ballases requires high carbon supersaturation.

Synthetic ballases have been crystallized both at high static pressures (high pressure—high-temperature—HPHT) [17,18] and using the Chemical Vapour Deposition approach (CVD) [19–21]. Detailed studies of the texture and microstructure of the HPHT ballases showed close similarities with those of starting graphite [22]. Together with studies of metal-based inclusions in these samples [23], these works showed that the HPHT ballas forms via graphite-to-diamond transformation during percolation of the metal melt into a graphite piece [24], thus making a direct comparison with natural material impossible. Nevertheless, high carbon supersaturation is a prerequisite for ballas synthesis.

In the case of CVD-grown samples, the formation of ballas-like deposits was observed in a broad range of growth medium compositions and substrate temperatures [20,21]. It was shown that in a single experiment, the formation of both high-quality diamond crystals and ballas deposits is possible with a formation of a continuous series from one end-member to the other. A qualitatively similar sequence is also plausible for natural diamonds since the size and relative volume of single crystalline domains in natural ballas vary widely.

Despite a long history of studies with variable degrees of detalisation, many essential features of the ballas structure and genesis remain debatable, complicating the reconstruction of their formation processes. In the current work, we present the results of a comprehensive investigation of two samples from Brazil, which, according to morphology and internal structure, belong to "classic" radial ballases.

## 2. Materials and Methods

Two ballas samples were collected in a placer of the St. Antonio River (belonging to the Rio de Prata river system), Minas Gerais state, Brazil [25]. Single-crystal diamonds are also abundant in the placer. The ballas samples are termed as "large" and "small"; the masses are 112 and 80 mg, respectively (Figure 1).

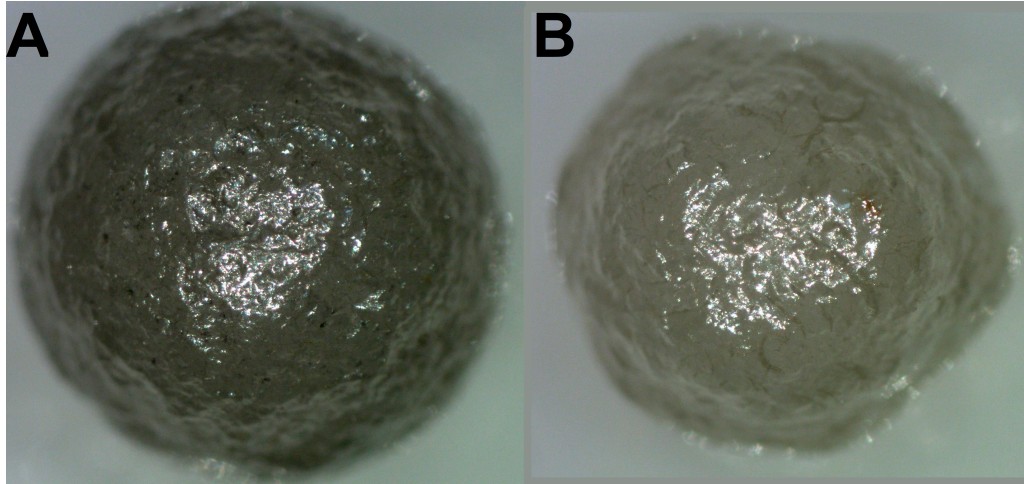

**Figure 1.** Optical images of the studied samples. (**A**)—"large" ballas, (**B**)—"small". The diameter of the "large" specimen is 4 mm, and the "small" one is 3.5 mm.

The morphology of the samples was studied using 3D optical microscopy (Keyence VHX-1000), Scanning Electron Microscopy (SEM, JSM-6510LV), and X-ray tomography. For the tomographic investigation, a homemade TOMAS system was employed [26]. It incorporates a Mo X-ray tube operating at an accelerating voltage of 40 kV and 40 mA current; data collection is performed using XIMEA-xiRay11 high-resolution X-ray detector (XIMEA, Marianka, Slovakia). This detector acquires images with a spatial resolution of 10 μm and a field of view of $36 \times 24$ mm. The samples were rotated relative to a fixed vertical axis, and 400 projections were measured with a step of 0.5 degrees and an exposure of 4 s per frame. The orientation of the comprising crystallites was assessed using an XGT-7200V X-ray fluorescence microscope equipped with an Rh tube operating at 1 mA and 50 kV accelerating; a 10 μm monocapillary was used.

In the second stage, the samples were cut with a $CO_2$ laser. Central plates 0.4–0.5 mm thick, with one side passing through the geometrical center of the diamonds, were made. Unfortunately, during the polishing, the plate from the "small" ballas was partly destroyed: a radial sector was torn out. The obtained plates were studied using color cathodoluminescence (CL), infrared (FTIR) microscopy, and photoluminescence. CL images were obtained with a Cameca MS-46 microprobe equipped with a high-resolution Videoscan-285 camera, which provides real color images [27]. The images were obtained at 20 keV and 20 nA at room temperature. For the FTIR investigation, a Nicolet iN10 microscope was used, the samples were measured in transmission mode with a 50 μm aperture, and at least 64 scans at 2 $cm^{-1}$ spectral resolution were acquired. Diametrical profiles passing through the centers of the samples were measured to assess radial variations in defect speciation and content. Photoluminescence (PL) spectra and maps were acquired at room temperature using a Renishaw inVia Reflex spectrometer equipped with a 405 nm laser. For the PL mapping, grids with a step size of 5 μm were used.

The carbon isotopic ratios were measured using a Flash EA 1112 (Thermo Fisher Scientific) coupled to a Finnigan Delta Plus XP isotope-ratio mass spectrometer. Small cut-off pieces of the outer parts of the ballas samples were treated in ultrapure nitric acid (65%, Merck, Germany) in an ultrasonic bath, rinsed in ultrapure water, and dried at 40 °C. Each sample was crushed in an agate mortar; individual fragments of 0.05–0.1 mg were inserted into Sn capsules. The combustion reactor operates at 1200 °C; the reduction reactor was maintained at 650 °C. All values of carbon isotopic composition are reported in standard δ notation in the Vienna Pee Dee Belemnite Reference standard (VPDB) scale ($\delta^{13}C$ VPDB). Seven fragments of each sample were analyzed. The accuracy of the isotopic data was evaluated by analyzing the certified reference material NBS 22 ($\delta^{13}C = -30.03 \pm 0.04‰$), IAEA, Vienna, Austria. High-purity helium (>99.9999%; NIIKM) and $CO_2$ (≥99.999%; Voessen, Moscow, Russia) were used as a carrier gas or working standard gas, respectively.

## 3. Results

### 3.1. Morphology

The external morphology of the samples is well visualized by X-ray tomography (Figure 2). The "large" specimen is closer to an ideal sphere (Figure 2A,B), whereas the "small" one possesses flattened "faces" (Figure 2C,D). Despite these differences, scanning electron and high-resolution 3D optical microscopies indicate general similarity of surface patterns. The surfaces of both samples consist of numerous flat stepped fragments of triangular/rhomboidal shape with pronounced edges (Figure 3). The surface features of the "large" specimen are more pronounced than those of the "small" one. The general geometrical motif reflects trigonal crystallite shapes, suggesting that most are octahedra. Cyclic twins are observed (Figure 3C) but in small numbers only. Fragments of crystallographic faces are arranged in clusters associated with macroscopic trigonal hills. The mutual arrangement of edges and apexes of the crystallites shows ordered rotational relative displacements. The rotation angle reaches 180° with a 30° step. In general, the morphology of the ballas surface implies the splitting of comprising crystallites.

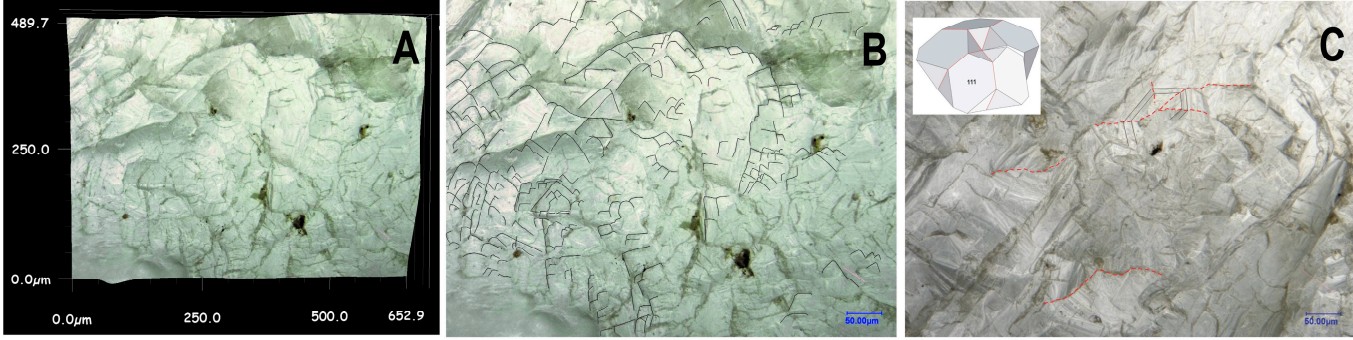

**Figure 2.** X-ray tomography (**A**,**C**) and sketch of the principal surface features (**B**,**D**). (**A**,**B**)—"large" ballas, (**C**,**D**)—"small".

**Figure 3.** 3D optical microscopy images of the "large" ballas. (**A**,**B**)—quasi-3D representation of the surface. Lines highlight trigonal motif of the surface features. (**C**)—rare cyclic twin, the inset shows its simplified scheme.

### 3.2. Structure

#### 3.2.1. General Structural Features

The radial fibrous structure of the samples is demonstrated in X-ray diffraction (Figure 4). The employed X-ray fluorescence microscope is equipped with several detectors. In the absence of filters, the incident X-ray beam consists of Rh characteristic lines superposed on a continuum. For some crystallographic planes, the Bragg diffraction condition may be fulfilled, and the diffracted beam reaches the detector. Scanning a sample with a narrow (10 µm) beam allows the construction of intensity maps of the reflections. In principle, knowledge of the diffraction peak energy and the scattering angle permits the identification of interplanar spacing corresponding to the reflection and, thus, indexing it. However, the scattering angle depends on the exact position of the diffracting volume relative to the detector and is generally unknown, especially for samples with non-planar geometry, such as the studied ballas samples. Subsequently, we refrain from indexing the reflections, but the intensity mapping shows that given sets of crystallographic planes are arranged in fibers.

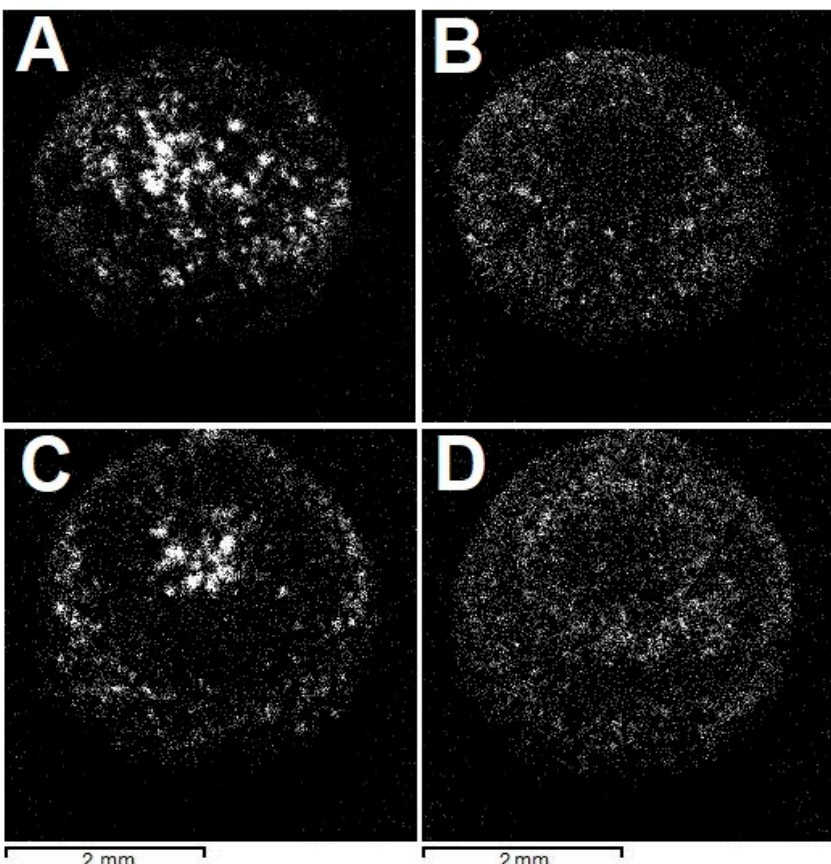

**Figure 4.** Spatial distribution of (unindexed) Bragg reflections, highlighting the radial fibrous structure of the studied ballases; see text for detail. (**A**,**B**)—"large" ballas, (**C**,**D**)—"small".

Grain orientation was investigated using EBSD. On the "large" sample, both the whole polished surface and a smaller part in the central region were analyzed; on the "small" ballas, only a relatively small central region was studied, but with a relatively high spatial resolution of 1.7 µm. Stereographic diagrams of both samples are qualitatively similar, showing well-defined clustering of reflections (Figure 5). Detailed EBSD mapping of a region of the "large" ballas reveals that reflections form an elongated cluster in the {110} plane. This observation is in line with the results of the analysis of the surface features, suggesting the rotational splitting of crystalline fibers comprising ballas and the presence of growth-related autodeformations arising from rotation and bending [28,29]. In the studied case, the rotation occurs in the {110} direction. The angular scatter of the reflections for

different planes is 56 and 69° for {100}, 59, 63, 76° for {110}, and 68, 72° for {111}. This rotation may explain zoning, revealed by cathodoluminescence; see below.

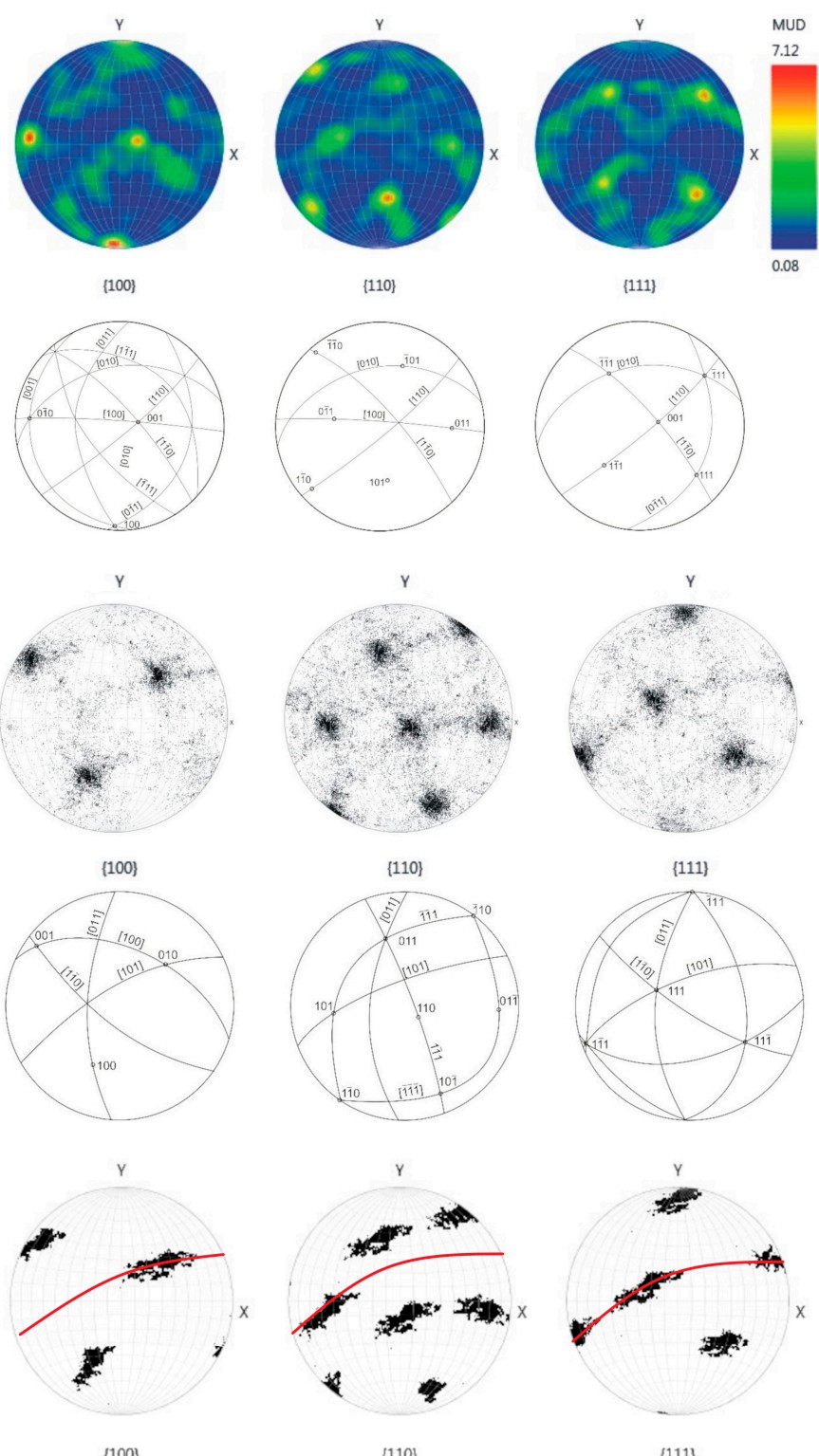

**Figure 5.** Stereographic diagrams and crystallographic interpretation of reflection positions on Schmidt plots, EBSD data. Positions of 100, 110, 111 reflections along {100}, {110}, {111} directions. Two upper rows—the central part of the "large" specimen; other rows correspond to the whole studied surface. The thin red line in the bottom row shows the {110} plane. Schemes show projections of crystallographic directions and clustering of the reflections.

### 3.2.2. Features of Cathodoluminescence Images

A prominent feature of the cathodoluminescence images of the central cross-sections of both samples is the presence of pervasive light blue "feather"-like formations surrounding visually uniform crystallites of various sizes and shapes (Figure 6A,C). The single crystalline domains, especially in the central zone, are often deformed as manifested by intense yellow-green luminescence caused by H3 defects (with a minor H4 component), decorating glide planes and dislocations. In diamonds, the {111} is the most common glide plane. Close inspection of the glide planes arrangement shows that single crystalline domains separated by only several tens of microns may be misoriented by up to 30 degrees (Figure 7A).

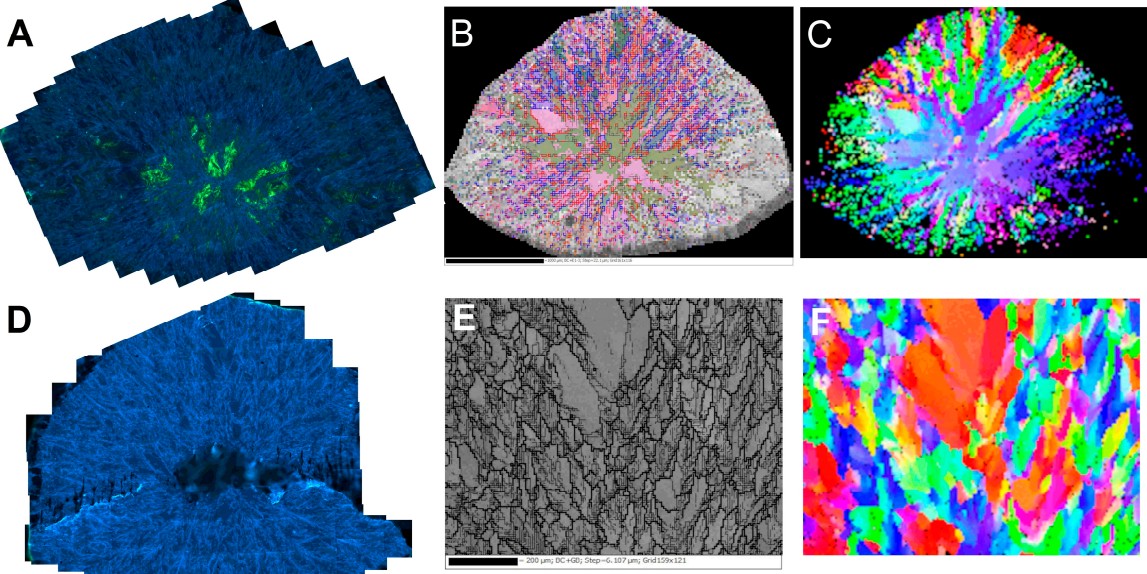

**Figure 6.** Cathodoluminescence (**A**,**D**) and EBSD maps. Top row—"large" specimen, bottom row—"small" one. (**B**,**E**)—grain and sub-grain (thin lines) boundaries of the studied domains. In B, the SEM image is superimposed on the Euler angles map of the grains' misorientation. (**C**,**F**)—orientation of the grains color-coded as Euler angles. The low quality of the sample edges of the images (**B**,**C**) is due to the technical limits of the movement range of the sample holder.

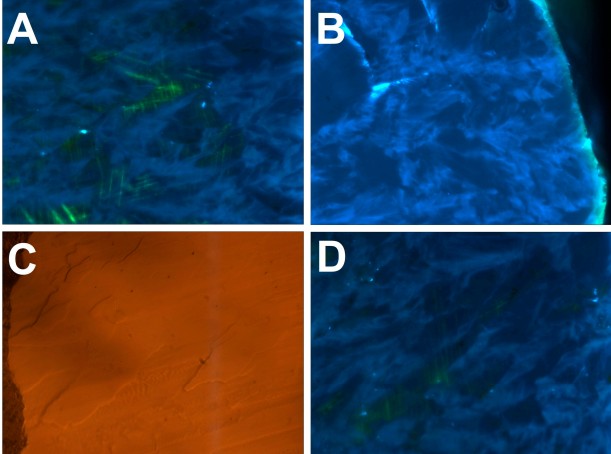

**Figure 7.** Details of cathodoluminescence (CL) and optical reflection images (**B**)—the "small" sample, other panels—the "large" one). Width of every image—300 μm. (**A**)—misoriented single crystals separated by the "feathers". The misorientation is visualized by the relative orientation of the glide planes. (**B**)—the "feathers" reaching the growth surface often give rise to hillocks. (**C**,**D**)—optical reflection and CL image of the same region, one can see that the "feathers" often show negative relief after the polishing.

As shown by photoluminescence mapping (Section 3.5), the "feathers" are visible on background broadband A-band luminescence due to enrichment in N3 nitrogen-related defects. They consist of a pure diamond without admixture of a foreign phase. It is possible to observe correlations between the surface relief of the polished surface and the "feathers": usually, they correspond to negative relief, implying more efficient removal of material during the polishing compared to single crystalline domains (Figure 7C,D). This is explained by the highly anisotropic hardness of the diamond and is a clear manifestation of relative crystallographic misorientation of the "feathers" and single crystalline domains.

The axes of the "feathers" are always directed from the ballas center towards the periphery. They may be macroscopically long; some of them can reach (almost) the full radius of a specimen, as can be seen on the "small" ballas. When a "feather" reaches the specimen surface, it often gives rise to a hillock (Figure 7B). Consequently, it is plausible that these formations represent diamond fibers characterized by non-crystallographic branching and making the ballas samples similar to compact spherulites [30].

Comparison of the spatial distribution of grain boundaries and Euler angles in the "large" ballas with the CL mosaics (Figure 6A–C) reveals that the "feather" features are made of smaller domains with considerable mutual misorientation; these formations surround relatively large domains with weakly pronounced (or even absent) internal disorder. Although the transition between the "feathers" and single crystalline domains is smooth, in many cases, misorientation between them often induces significant strain, as shown by the appearance of glide planes in the latter domains (Figure 6A). Note that the glide planes decorated by H3 defects never cut the surrounding "feathers", as would be expected for a post-growth deformation caused by external forces. The appearance of the glide planes implies deformation in a brittle regime. At the same time, the studied ballases were clearly formed at high temperatures. The brittle-to-ductile transition in diamond was studied in [31], and it was shown that glide plane formation might indeed occur at high temperatures, providing sufficiently high confining pressures. These facts support the hypothesis about the occurrence of the autodeformation process during the ballas formation.

Closer examination of the CL mosaics of the "large" ballas reveals the presence of concentric zonation: in the central part ~1 mm in diameter, the density of the "feathers" is considerably higher than in the outer part. The single crystalline domains in the central part are few, relatively large, and heavily deformed. The transition between the zones is not reflected in IR spectra (Section 3.4). Zonation observed in spherulites of various substances is usually interpreted as the rotation of the constituent crystallites (e.g., [30]). As discussed above, the fibers comprising the studied samples rotate along [110]. Presumably, the observed zoning of the "large" ballas is also explained by this phenomenon.

Very similar structures were previously observed in the TEM study of CVD-grown ballas [32], where they also surround high-quality diamond single crystallites. In [32], the "feathers" were interpreted as branched twin agglomerations; this assignment is based on a paper [33], which, however, does not contain supportive data. Although this interpretation might seem reasonable, it is not supported by direct measurements, such as electron diffraction. Interestingly, in EBSD analysis of misorientations of neighboring grains, one can find angles close to those expected for highly symmetric boundaries (CSL, Coherent Site Lattice), such as $\Sigma19$ ($\Theta = 26.5°$) or $\Sigma13$. However, analysis of the whole grain population indicates that such boundaries are rare. Moreover, even the most energetically favorable $\Sigma3$ ($\Theta = 70.5°$) boundaries are almost absent. Contact and cyclic twins are observed on the surface of the "large" ballas using optical microscopy (Figure 3), but their number is insignificant. Combined with the stereographic distribution of reflections, this indicates that although twinning is present in the studied samples, it is not widespread. In [32], it was shown that the "feather"-like twin agglomerations are limited to micron-size domains. If correct, this would imply that the identification of the twins using EBSD may depend on the step of the measurement grid. In [18], the twinning was assumed from characteristic surface features of synthetic ballas samples. At the same time, a fundamental X-ray study

of natural ballases [7] failed to detect twins. Therefore, the importance of twinning for the formation of ballas remains unclear.

*3.3. Inclusions*

In IR spectra of some spots in the "large" specimen, especially in its peripheral part, absorption bands with maxima at 865, 878, 1063, and 1090 $cm^{-1}$ are observed. These bands are related to microinclusions of silicates and micas, e.g., phlogopite. Similar phases and quartz were earlier reported for ballas [12] and ballas-like [11] diamonds; abundant quartz particles were found using optical microscopy after the oxidation of Brazilian ballases [7].

Optical microscopy reveals the presence of numerous inclusions a few microns in size. Unfortunately, attempts to identify individual mineral phases using Raman spectroscopy were unsuccessful, mainly due to the strong background luminescence of the diamond matrix. The X-ray radiography and tomography did not detect the microinclusions due to sizes considerably smaller than the pixel size (~10 μm) of the detectors employed. Note that in our previous studies of ballases and ballas-like diamonds, microinclusions were located by the later methods and are preferentially encountered on intergrain boundaries [11] or cracks [34]; similar observations were made in [7].

Transmission electron microscopy (TEM) was employed to investigate nanosized inclusions in foils extracted by Focussed Ion Beam (FIB). After ion beam polishing, the thickness of the foils was less than 100 nm. In the foil extracted from the "small" ballas, numerous inclusions 10–15 nm in size were found (Figure 8). All attempts to measure chemical composition using EDX spectroscopy failed, implying that the inclusions contain only light elements such as C, O, N, and H. In most cases, the inclusions are arranged in chains, suggesting their association with intergrain boundaries. Note that the spatial distribution of the inclusions is highly heterogeneous, even on the scale of foil, and their volume fraction is small. The absence of noticeable absorption bands of $CO_2$ and/or $H_2O$ in IR spectra also suggests insignificant content of these phases, if any.

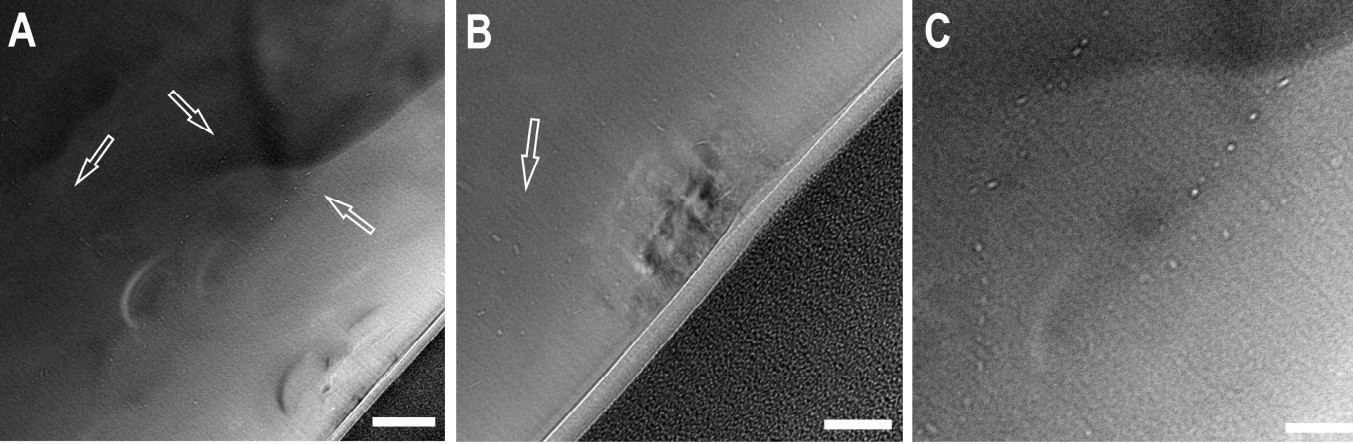

**Figure 8.** Transmission Electron Microscopy images of a foil from the "large" ballas. Small bright spots (in (**A**,**B**) pointed by arrows)—inclusions, possibly fluid. Scale bar—for (**A**)—200 μm, for (**B**,**C**)—100 μm. The black region in the bottom right part — Pt protective layer used for sample preparation by FIB.

*3.4. Infra-Red Spectroscopy*

Infra-red absorption spectra show significant differences between the samples both in total nitrogen content and in their aggregation state (Figure 9). In the "small" specimen, the nitrogen concentration in the central part is ~560 at.ppm (21 ppm as A-defects and 540 ppm in B-defects); in the outer region, it increases to 706 at.ppm (21 ppm in the A-form, 685 ppm in the B-form), but the aggregation degree (%B) changes only marginally from ~98 to 97%. In this specimen, the platelet defects are also present; the peak maximum (1362 $cm^{-1}$) and intensity (~7 $cm^{-1}$) do not change along the radial profile. In the "large" ballas, the

total nitrogen concentration is considerably lower: 170 at.ppm in the center (44 ppm as A, 127 ppm as B-defects) and 190 ppm (50 ppm as A, 140 ppm as B-defects) at the edge. The aggregation degree is also virtually constant—34–35%. For both samples, the increase of the nitrogen content in the radial direction appears to occur gradually. However, the relatively large thickness of the central plates (0.4–0.5 mm) and spherical morphology of the ballas samples leads to the inevitable integration of the absorption over adjacent zones; the effect is especially pronounced in the peripheral part. Analysis of the time-temperature history of the samples indicates fairly high annealing temperatures: assuming the annealing duration of 1.5 Ga, the temperature for the "small" sample is just slightly less than 1250 °C and for the "large" is 1125 °C (Figure 9B). For both samples, some temperature decrease in a radial direction is plausible. Note that despite relatively high annealing temperatures, the "feather"-like features do not show evidence of recrystallization.

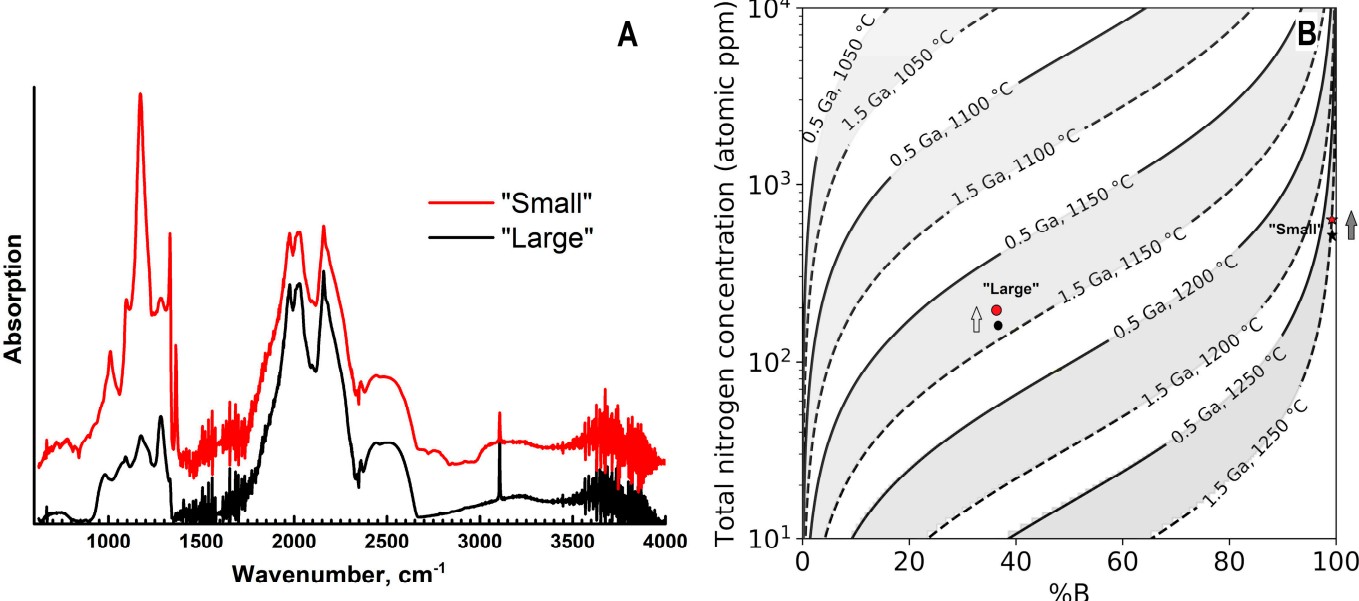

**Figure 9.** Infra-red spectra of two samples (**A**) and time-temperature history inferred from the nitrogen aggregation state (**B**). In A, the curves are displaced vertically for clarity. In (**B**), arrows indicate the direction from the sample center to the edge.

In both samples, hydrogen-related absorption lines at 1405, 3107, and 4496 cm$^{-1}$ are present. The intensity of the 3107 cm$^{-1}$ line is constant in the sample cross-section, and its values are ~3.5 cm$^{-1}$ in the "large" and ~1.5 cm$^{-1}$ in the "small" ballas. Absorption lines related to the inclusions of other phases are discussed in the preceding Section 3.3.

### 3.5. Photoluminescence Spectroscopy

Photoluminescence spectra of both samples can be considered as a superposition of two main types of defects: (1) a combination of N3 and H3 centers in comparable amounts, and (2) a strong domination of the H4 center (Figure 10A). The luminescence of the main volume of both samples is determined by the N3 (zero phonon line, ZPL, at 415 nm) and the H3 (ZPL at 503 nm) defects and their phonon replicas; the spectra of the second type are encountered only in some peripheral parts of the samples.

The spectra of the main volume are complicated by several other bands peaked at ~452, 464, 480, and 485 nm. Unique assignment of these bands is difficult due to overlap with the N3 replicas. However, variations of the shape, relative intensity, and likely presence of additional peak components at 452 nm suggest the presence of yet another defect. It might correspond to the so-called "yellow band" observed earlier in natural brown diamonds [35]. The large width of this band may indicate significant stresses in the diamond lattice.

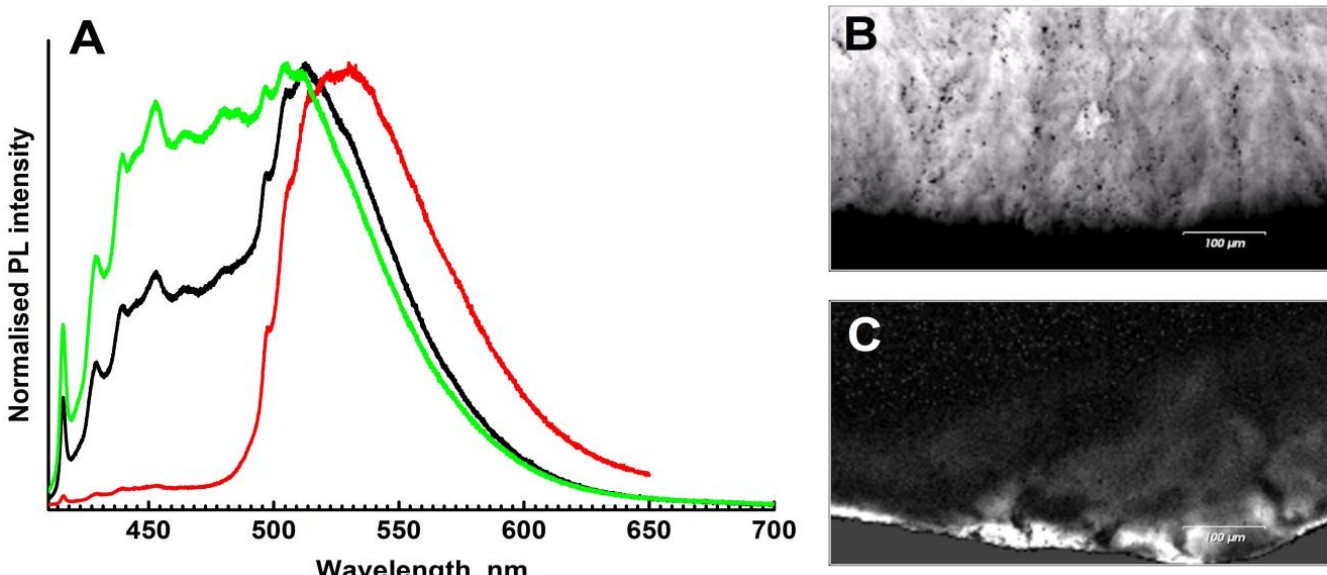

**Figure 10.** Photoluminescence spectra and maps, $\lambda_{exc}$ = 405 nm. (**A**)—representative spectra of different spots; intensities are normalized to the [0, 1] interval. (**B**)—map of the N3 defect distribution (spectral range 414–419 nm); (**C**)—distribution of the H4 defect.

Photoluminescence mapping shows that the "feather" formations discussed above are enriched in the N3 defect (Figure 10B); actually, it is the luminescence of this center which makes the "feathers" visible in CL images. Note, however, that not only differences in N3 concentrations but also variations in the A-defect content may be important since the A defect may effectively quench the N3 luminescence. Although in natural diamonds the growth sector dependence of the nitrogen content is not as pronounced as in HPHT synthetics, the diamond fibers responsible for the "feathers" may have incorporated nitrogen at a different rate compared to single crystalline domains.

The PL mapping also shows that the H4 defects (complex of the nitrogen-related B-defect and vacancy) are mostly confined to peripheral parts of ballases; these defects are responsible for the narrow green-yellowish rim in CL mosaics (Figure 10C). In these regions, the luminescence of the H4 defects is markedly higher than that of N3+H3. Since, for all these centers, the excitation efficiencies with a 405 nm laser (energy 3.06 eV) are comparable, it is likely that the H4 defects are indeed more abundant. Agglomeration of the H4 defects in the outer regions of the samples is not very common and is likely explained by mechanical deformation in the course of interaction with surrounding minerals. The influence of radioactive minerals/fluids is less plausible since it is not easy to reconcile with the high temperatures required for the H4 defect formation. Note that pleochroic rings surrounding trapped radioactive minerals were earlier reported for ballas-like diamonds [11]. In any case, the formation of the H4 defects in the outer parts indicates prolonged high-temperature annealing already after the termination of the ballas crystallization.

### 3.6. Carbon Isotope Composition

The obtained results of carbon isotopic composition are presented in Table 1. Both samples are characterized by $\delta^{13}C$ values (−5.42 and −7.11‰) typical for juvenile mantle carbon (e.g., [36]).

**Table 1.** Carbon isotopic compositions of the studied ballases ($\delta^{13}$C VPDB, ‰).

| Subsample | "Large" Ballas | "Small" Ballas |
|---|---|---|
| 1 | −4.16 | −8.53 |
| 2 | −6.29 | −5.91 |
| 3 | −5.16 | −8.50 |
| 4 | −5.55 | −5.92 |
| 5 | −4.28 | −7.99 |
| 6 | −6.61 | −6.45 |
| 7 | −5.89 | −6.50 |
| Mean (±1σ) | −5.42 (±0.95) | −7.11 (±1.18) |

## 4. Discussion

Ballas samples studied in our work represent classic samples with radial fibrous structures. The main volume of the samples is filled with fibrous crystallites with fan-shaped branches facing the outer surface of the sample. EBSD results indicate the dominance of the [110] texture in the studied samples, in agreement with earlier studies of natural ballases, ballas-like and spherocrystal diamonds [7,15]. This texture may develop by the propagation of growth layers nucleated on cubooctahedral diamond crystallites [19]. However, other texture directions—{111} and {100} were established as well [15,37].

A significant new result of the current work is the observation of rotational splitting of comprising crystallites, manifested in the mutual orientation of surface features (Figure 3) and the distribution of reflection spots on stereographic projections (Figure 5). The splitting is also beautifully manifested in peculiar structural details revealed by luminescence imaging—the "feather"-like features containing a higher concentration of nitrogen-related N3 defects and, as suggested by EBSD grain mapping, most likely representing branching diamond fibers. The spatial distribution of "feathers" in the "large" specimen shows clear zoning: they are much more abundant in the central part (Figure 6). This resembles banding known for some spherulites [30] and is likely related to the rotation of the comprising fibers and related distribution of strain fields.

The reasons for the fibers' rotation are unclear and might be related to the influence of impurities; see review in [30]. However, according to IR spectroscopy, radial variations of the principal chemical impurity in diamond—nitrogen—occur gradually. However, as discussed in Section 3.4, due to technical reasons, the fine zoning of the N content might elude detection. Therefore, although the zoning observed in the CL image of the "large" ballas is not pronounced in the infra-red spectra, we cannot fully exclude changes in N content associated with the zoning. At the same time, the "small" specimen is also characterized by a gradual increase of the N content towards its rim, but no zoning is observed. Earlier [9,12], it was suggested that nitrogen concentration and its aggregation state are positively correlated with the volume fraction and development of single crystalline domains in ballas. In our work, two samples with considerably different but significant amounts of nitrogen are investigated, and no relationship with the crystallinity is found. The observed N concentrations are not unusual for ordinary single crystals. Consequently, it is unlikely that the nitrogen impurity has triggered the formation of the fibers and/or their rotation.

Examination of nitrogen content and speciation suggests high formation temperatures in the range of 1125–1250 °C, consistent with growth in the lithospheric mantle. In addition, the presence of the H4 defects in the peripheral part of the ballas samples implies post-growth high-temperature deformation processes. The investigation of the internal structure of natural ballas diamonds using TEM revealed a polygonisation of dislocations, also indicating deformation at high temperatures [13]. Comparison with literature data [9,11,12] shows that nitrogen content and inferred formation temperatures may vary widely for ballas diamonds precluding a single restricted origin.

The $\delta^{13}C$ values for the studied ballas samples ($-5.42 \pm 0.95‰$ and $-7.11 \pm 1.18‰$) belong to the main mode of both peridotitic and eclogitic diamonds spanning the range from $-2‰$ to $-8‰$ [36] or from $-1‰$ to $-9‰$ [38]. Only a few values of ballas $\delta^{13}C$ have been reported. Ballases from the Sayan placers in Russia have $\delta^{13}C$ of $-14.2$, $-19.0$, and $-21.4‰$ [10]. These isotopically "light" values are similar to eclogitic diamonds and possibly imply the involvement of recycled sedimentary carbon. On the other hand, ballases from the Sytikanskaya kimberlite pipe in Yakutia are characterized by heavier carbon: $\delta^{13}C$ is $-8.7$ and $-10.1‰$; ballases from the Vilyuy River placer deposit possess $\delta^{13}C = -12.9‰$ [5]. In [11], the following values of $\delta^{13}C$ were reported for ballas and ballas-like diamonds from Brazil: $-5.35$, $-11.2$, $-6.2$, $-6.33‰$; the three later values are related to diamonds with pronounced ballas texture in X-ray Laue patterns. Taken altogether, this dataset shows rather significant variations in the carbon isotopic composition of ballases from different sources.

## 5. Formation of Ballas Diamond

The data obtained allow us to make some suggestions on the origin of ballas. The revealed regularities in the structure of the studied samples can be described as compact spherulites consisting of fibrous twisted diamond crystallites characterized by non-crystallographic branching, which engulf single crystalline domains. The main volume of the samples is filled with fibrous crystallites with fan-shaped branches facing the outer surface of the sample.

Radial fibrous ballas may be formed both on the nanocrystalline diamond core (HPHT experiment, [39]) and on the single crystal diamond core (natural samples [15,16,40]). Experiments on CVD synthesis of ballas and ballas-like diamonds showed that their formation is possible in a wide range of the growth medium chemical composition ($CH_4/H_2$ ratio) and temperature [19–21]. High-quality single crystals and ballas can be obtained in a single experiment. The latter variety forms in regions of low plasma intensity, which, in particular, corresponds to less efficient ionization of species and etching of non-diamond carbon. These factors influence the kinetic coefficient, i.e., efficiency (and rate) of carbon incorporation into a growing diamond.

Presumably, the kinetic coefficient plays the principal role in forming the unusual radial fibrous structure of ballas. In general, the spherulite formation is favored by a combination of a small kinetic coefficient and high crystallization driving force caused by high supersaturation/supercooling [30]. The model of the formation of ballas and ballas-like diamonds based on investigating several natural diamonds [16] fits this scenario. A high concentration of impurities on the growth surface may prevent the motion of the growth steps (poisoning effect), but as shown above, nitrogen impurity is an unlikely contributor by itself. However, speciation of carbon and, possibly, of nitrogen, in the growth medium appears to be important for the onset of radial growth.

Remarkably, carbon isotopic composition, nitrogen concentration, and speciation, as well as inferred growth temperatures, fall into the ranges established for diamonds from kimberlites. Consequently, the rarity of ballas diamonds is not related to a novel type of parent rocks but reflects peculiar conditions in a growth chamber.

## 6. Conclusions

The results of a comprehensive study of two ballases from Brazil allow us to draw some conclusions about the structure and genesis of this peculiar diamond variety.

1. Ballas diamonds can be considered spherulites, known for many inorganic and organic substances. Radial fibers surround single crystalline domains of various sizes. Twinning is not pronounced in the studied samples. Diamond fibers radiating from the specimen center towards the periphery show features of rotational splitting. The rotation may lead to the appearance of zoning in ballases.

2. Pervasive "feather"-like formations observable in cathodoluminescence are a remarkable structural feature of ballases. They are likely related to the diamond fibers,

undergoing intense splitting and rotation during ballas formation. These features are enriched in N3 nitrogen-related defects.

3. Nitrogen concentration in studied ballases varies in a broad range. It gradually increases in a radial direction, but the aggregation degree remains virtually constant. No correlation between N content and/or aggregation with the relative volume of single crystalline domains is found. The aggregation degree may reach 96–97%, implying formation temperatures up to 1250 °C with a gradual decrease towards the end of the growth. Principal luminescing defects are represented by N3, H3 defects, and some other bands. In the outer part, the H4 defect is present, indicating deformation and prolonged annealing of already formed ballas samples

4. The studied ballases are formed at high temperatures (1125–1250 °C), which corresponds to the position of the mantle adiabat at a pressure of at least 5 GPa, i.e., are in the domain of formation of single-crystal diamonds in kimberlites. The carbon isotopic composition of the studied ballases ($\delta^{13}$C = −5.42, −7.11‰) belongs to the main mode of mantle-derived diamonds.

5. Ballas diamonds were formed in conditions of high supersaturation and low kinetic coefficient. The role of nitrogen as a trigger for the fibers' formation appears to be minor; speciation of carbon in the growth medium might be more important.

**Author Contributions:** Conceptualization: A.A.S. and F.V.K.; methodology: A.A.S.; formal analysis: A.A.S., F.V.K., A.D.P., V.O.Y.; investigation: A.A.S., F.V.K., V.O.Y., D.A.Z., A.A.A., O.M.Z., M.S.N., O.V.K.; writing—original draft preparation: A.A.S., F.V.K., A.D.P.; writing—review and editing: A.A.S., F.V.K., A.D.P.; visualization: A.D.P., V.O.Y., D.A.Z. All authors have read and agreed to the published version of the manuscript.

**Funding:** In part of morphology investigation, this research was funded by the state assignments of DPMGI SB RAS. This work was performed within the State assignment of Federal Scientific Research Center "*Crystallography and Photonics*" of the Russian Academy of Sciences in part of X-ray microtomography measurements. XRF and spectral measurements were performed using equipment of CKP FMI IPCE RAS. CL and 3D microscopy were performed with the support of the State assignment of IGEM RAS. Felix Kaminsky and Olga Kuznetsova are contributors to the Project of the State Assignment of Vernadsky Institute of Geochemistry and Analytical Chemistry, Russian Academy of Sciences; FMUS-2019-0013.

**Data Availability Statement:** Analytical data will be provided upon reasonable request to the corresponding author.

**Acknowledgments:** We appreciate the useful comments of two anonymous reviewers.

**Conflicts of Interest:** The authors declare no conflict of interest.

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
