# Peer review of "Structural Peculiarities of Natural Ballas—Spheroidal Variety of Polycrystalline Diamond"

_crystals, doi:10.3390/cryst13040624_

Round 1
Reviewer 1 Report
The paper documents detailed study of two ballas diamonds from Brazil. The main limitation is the small number of samples studied, but these are documented in detail with various spectroscopic and crystallographic techniques. Minor comments and edits, largely relating to English usage are tracked in the uploaded document. Unfortunately the tracked changes have caused a problem with the autogenerated line numbers but this should be easily resolved as the suggested changes are clear.

Author Response
Dear Dr. Liu,
Please find attached revised version of manuscript “Structural peculiarities of natural ballas – spheroidal variety of polycrystalline diamond” by Shiryaev and co-authors.
We highly appreciate attentive consideration of initial version of the manuscript by two reviewers. The absolute majority of suggested corrections is implemented.
New information/clarifications requested by the reviewers are highlighted in yellow.
Several points raised by the reviewers require detailed replies.
1) Discussion of the microscale twinning proposed in Ref. 32 (Joksch et al, 1994). The authors of the (really good and useful) paper [32] refer to another paper in discussion of an important issue of presence of microscale twins; let’s cite the corresponding place from ref. 32:
“The twin agglomerations started from a single point and branched frequently, thus leading to the spherulitic growth of ballas [24].”
The cited paper [24] is indeed difficult to find. We have managed to get a copy and were somewhat surprised to find that it is a review of graphitic precipitates in cast iron. Graphitic spherulites in cast iron might indeed show twinning, but there is no obvious connection of the graphite in cast iron and ballas diamond.
Anyway, we have rewritten the sentence in a polite way as requested.
2) In the initial version of the manuscript it was stated that indexing of X-ray reflections from our XRF measurements is impossible. This point was criticized by the Reviewer 2 and thus requires explanation.
The measurement geometry of the employed spectrometer is similar to backscattering scheme of X-ray diffraction, which is well-known. An important difference of this setup with a conventional diffractometer is that polychromatic radiation is used. For a given d-spacing of the crystalline material and the given scattering angle Θ of the instrument radiation with energy given by formula E = 1,242/2d sin Θ will be scattered. Consequently, if one knows the energy of a reflection AND the scattering angle, the interplanar spacing may be calculated, thus allowing the indexing (e.g., monograph ‘Laboratory Micro-X-Ray Fluorescence Spectroscopy” by Michael Haschke, Springer, 2014). Some papers (e.g., doi:10.1107/S1600576714000314) describe the mapping of the orientation of the crystallites using an XRF spectrometer, but in such cases that scattering angle is known and fixed.
However, in our case the scattering angle is difficult to measure reliably due to peculiarities of the measurement geometry. In particular, for a sample with highly irregular morphology it is impossible to indicate the distance to the detector and thus the scattering angle is poorly defined. We thus refrain from indexing of the reflections due to ambiguities.
We hope that the revised version will be suitable for publication.
On behalf of all co-authors,
A.Shiryaev

Reviewer 2 Report
The authors of the proposed manuscript studied and structure and origin of the so-called ballases, a particular variety of diamond.
The article is very well written and organized and the results and presented in a designed scientific format. It is very well-cited and the quality of the figure is excellent. The description of the figures is also very good.
My only concern is the following: the authors often refer to unindexed Bragg reflections. Being a researcher working in the field of X-ray diffraction, the idea of not indexing a peak to a reflection is, in fact, strange. Nevertheless, I understand the reason. It is because of the high number of truncation rods observed in the spatial distribution (figure 4). Nevertheless, in order to accept the manuscript, the authors need to find literature and cite it where the Bragg peaks were not also indexed or to describe exhaustively the reason(s) to not index any Bragg reflection (preferentially with citations from known researchers in the field of crystallography).
The conclusions are also not presented in a common structure, at least according to my experience. Usually, conclusions are written as a continuous text where each major point (conclusion/idea) of the entire manuscript is summarized.
This last point is optional. The one concerning the Bragg peaks indexation is crucial. Therefore, the manuscript is suggested to be accepted only after the above point is clearly addressed. Decision: minor revisions are required.
Author Response

(The authors gave the same response as above.)
